# Targeting the CD47-SIRPα Axis: Present Therapies and the Future for Cutaneous T-cell Lymphoma

**DOI:** 10.3390/cells11223591

**Published:** 2022-11-13

**Authors:** Amy Xiao, Oleg E. Akilov

**Affiliations:** 1University of Pittsburgh School of Medicine, Pittsburgh, PA 15261, USA; 2Cutaneous Lymphoma Program, University of Pittsburgh, Pittsburgh, PA 15261, USA

**Keywords:** CD47, SIRPα, immunotherapy, mycosis fungoides, Sézary syndrome

## Abstract

The loss of CD47 on aging cells serves as a signal to macrophages to eliminate the target. Therefore, CD47 is a “do-not-eat-me” sign preventing macrophagal phagocytosis via interaction with its ligand SIRPα. Malignant lymphocytes of mycosis fungoides and Sézary syndrome express CD47 highly, thus, being ideal candidates for targeted anti-CD47 therapies. The classes of current anti-CD47-SIRPα therapeutic molecules present in a large variety and include monoclonal antibodies against CD47 and SIRPα, bioengineered SIRPα proteins, miRNAs, and bispecific antibodies. We provided a detailed analysis of all available investigational drugs in a contest of cutaneous T-cell lymphoma. A combination of blockade of the CD47-SIRPα axis and secondary targets in the tumor microenvironment (TME) may improve the clinical efficacy of current immunotherapeutic approaches. We evaluated the possible combination and outlined the most promising one.

## 1. Introduction

CD47 is a marker of self on all normal cells known to regulate cell turnovers, such as cell migration, cytokine production, and T cell activation [1,2,3,4,5,6]. Recently, much interest was attracted to the role of CD47 as a regulator of innate immune surveillance when bound to a membrane protein called SIRPα (SHPS-1/BIT/CD172a) on macrophages and other myeloid cells [7]. Phagocytosis is downregulated when SIRPα on a phagocyte binds with CD47 of the target cell [8]. Blocking CD47 makes it unable to restrain SIRPα, triggering phagocytosis [8] and creating an attractive therapeutic target. 

Many hematologic and solid tumors overexpress CD47, including acute and chronic myeloid leukemia (AML and CML) [9], acute lymphoblastic anemia [10], non-Hodgkin’s lymphoma (NHL) [11], multiple myeloma (MM) [12], and in solid cancers such as bladder, prostate, ovarian, lung, kidney, stomach cancers, hepatocellular carcinoma, gliomas, glioblastoma multiforme [13]. Cutaneous T-cell lymphoma (CTCL) is not an exception; moreover, significant overexpression of CD47 in Sézary syndrome [14] and mycosis fungoides [15] makes CTLC an ideal candidate for anti-CD47 therapy. High CD47 expression in MF correlates with worse outcomes [15]; CTCL tumors with higher CD47 levels have been shown to grow more rapidly and aggressively than their CD47 KO counterparts [15]. Furthermore, being very immunosensitive, CTLC is a desirable target for biologic therapies, and anti-CD47 is no exception. 

## 2. Anti-CD47-SIRPα Agents

All biologic agents targeting the CD47-SIRPα axis can be divided into five groups: monoclonal antibodies directed against CD47 and SIRPα, SIRPα proteins, small molecules, and bispecific antibodies (Table 1 and Table 2). 

### 2.1. Anti-CD47 Antibodies

Anti-CD47 antibodies endow the antineoplastic effect in three ways. First, they enable phagocytic uptake of tumor cells by antigen-presenting cells, leading to subsequent antigen presentation to CD4+ and CD8+ T cells, stimulating the anti-tumor adaptive immune response [16]. Second, they eliminate tumor cells via NK cell-mediated antibody-dependent cytotoxicity and complement-dependent cytotoxicity [17]. Third, they stimulate the apoptosis of tumor cells through a caspase-independent mechanism [18]. 

All the monoclonal antibodies against CD47 have an IgG4 Fc portion, except for AO-176, which contains IgG2 Fc [19]. IgG2 is predominantly responsible for IgG responses against bacterial capsular polysaccharides. Physiologically, IgG2 only significantly binds one Fcγ receptor, FcγRIIa. This offers an advantage of IgG2 over IgG4, as it does not bind to the inhibitory FcγRIIB receptor as IgG4 does. However, IgG2 weakly activates complement, while IgG4 does not. This is suboptimal as complement activation in the TME enhances tumor growth and increases metastasis [20]. IgG2-based anti-CD47 antibodies such as AO-176 may not be as effective as monotherapy. In a phase 1/2 first-in-human study of AO-176, 7/27 (26%) patients with acute myeloid leukemia had stable disease as the best response [21]. 

IgG4 antibodies (in all other anti-CD47 monoclonal antibodies) bind mainly to FcγRI and with less affinity to FcγRIIA, FcγRIIB, and FcγIIIA [22]. FcγRI is a very effective receptor for antigen presentation. FcγRI binding by weak immunogens leads to enhanced humoral immunologic responses. Conversely, the inhibitory role of FcγRIIB may be disadvantageous for tumor eradication; when engaged with antigen-bound IgG, FcγRIIB inhibits both antibody-dependent cellular cytotoxicity and phagocytosis [23]. Lastly, FcγIIIA is expressed in NK cells and plays a crucial role in the antigen-dependent cellular cytotoxicity [24]. This may be responsible for infusion reaction side effects.

Magrolimab has the most robust responses as monotherapy, with the highest ORR and CR rate as monotherapy. In a phase 1b trial for 43 AML patients treated with magrolimab, an ORR of 63% and 42% had complete remission. The median duration of response was 9.6 months, and the median time to response was 1.95 months [25]. In phase 1 preliminary data for SRF231, there were no complete or partial responders per RECIST criteria. However, a prolonged stable disease was observed (NCT03512340). There is no preliminary data yet on SHR-163 and TI-061. CC-9002 does not have an encouraging profile as monotherapy, so it is now being tested in combination with Rituxan, with results pending (NCT02367196). 

Given the ubiquitous expression of CD47, mAbs targeting CD47 have a high “drug sink” on erythrocytes, platelets, and other CD47-expressing cells, leading to anemia and thrombocytopenia [26]. The absence of CD47 on RBC marks it as senescent to macrophages, which facilitates its removal [27]. Thus, anemia is one of the main side effects of anti-CD47 therapy observed in 89% of patients in a phase 1 trial of magrolimab in patients with acute myelocytic leukemia (AML) [28,29]. This anemia is transient and reversible, with no observed grade 4 or 5 adverse events [28]. Interestingly, the combination of magrolimab with a demethylating agent, azacytidine, decreased the degree of anemia [28]. While AO-176 binds to RBCs negligibly, it still causes anemia in 22% of patients [19]. TJC4 should have unique RBC-sparing properties and binds negligibly to RBCs due to its novel conformational epitope, but no data are available yet since the clinical trials are still ongoing [30]. SRF231 and BI 867063 have shown no anemia, according to preliminary reports so far.

Other side effects, such as thrombocytopenia and infusion-related reactions associated with magrolimab, may be explained by the fact that magrolimab is an IgG4 mAb that binds to FcγRIIA and FcγRIIIa [31]. FcγRIIa triggers platelet activation and aggregation, and FcγRIIIa stimulates antibody-dependent cellular cytotoxicity, which may contribute to thrombocytopenia and symptoms of an infusion-related reaction, respectively [32,33].

Additionally, targeting CD47 with pure anti-CD47 mAbs may be disadvantageous due to a decrease in the antitumor effect secondary to nonspecific blocking of SIRPγ and SIRPα. SIRPγ expresses on T-cells. The interaction of SIRPγ with CD47 facilitates cell–cell adhesion, T cell transendothelial migration, and support of T cell co-stimulation with DCs [34,35]. Blocking SIRPγ with anti-CD47 mAbs, thus, reduces T cell activation, proliferation, and transmigration across the human endothelium (Figure 1). 

Several approaches such as anti-SIRPα antibodies, SIRPα -IgG fusion proteins, high-affinity variants, bispecific antibodies, and newer techniques are being developed to overcome these limitations.

### 2.2. Anti-SIRPα Antibodies: cc-95251 and BI 765063

If targeting CD47 causes thrombocytopenia and anemia, what about targeting SIRPα? SIRPα is predominantly expressed on myeloid cells and has restricted tissue expression, so it has been hypothesized that targeting SIRPα might overcome the abovementioned obstacles. Additionally, selective SIRPα blockade may avoid targeting SIRPγ, reducing the resistance to immunotherapy and potentially leading to better clinical response.

Preclinical data demonstrated that selective SIRPα blockade stimulated T-cell recruitment into the tumors by restoring macrophage chemokine secretion, promoting tumor-antigen cross-presentation. Anti-SIRPα monotherapy was effective in inhibiting tumor growth and preventing metastasis in highly enriched myeloid cells, triple-negative breast cancer, and mesothelioma. Thus, anti-SIRPα therapy should work better in other myeloid-enriched tumors such as CTCL.

Two anti-SIRPα monoclonal antibodies are currently in clinical trials: cc-95251 and BI 765063. Phase 1 clinical trials of cc-95251 alone and in combination with cetuximab and rituximab for advanced solid and hematologic cancers is in the recruitment phase (NCT03783403, NCT03783403), but BI 765063 was recently found to have clinical benefit as monotherapy in an ongoing phase 1 clinical trial for solid malignancies. In patients with various advanced solid tumors (ovarian (9), colorectal (8), lung (5), breast (4), melanoma (3), and kidney (3)), IV BI 765063 showed clinical benefit in 21/47 (45%) patients. It also increased PD-L1 expression in tumor cells two weeks after the first dose. While the clinical trial of BI 765063 is still ongoing, there have been no reports of anemia or thrombocytopenia. The most frequent side effects were infusion-related may be due to the exact mechanism observed with magrolimab since both mAbs are IgG4. The IgG4 binds to FcγIIIa, causing cytokine release, which may lead to symptoms of an infusion-related reaction. 

To complicate the story, the SIRPα has two variants: 1 and 2. Depending on which subvariant of SIRPα is being targeted, the mechanism of action of those antibodies and the effect on stimulation of phagocytosis maybe, to some extent different. It was reported that SIRP-1 antibodies induce macrophage internalization of SIRPα, reducing the availability of SIRPα to bind to tumor cells. SIRP-2 antibodies reduce CD47/SIRPα interaction via conformational change, producing reduced clustering of SIRPα and impeding interaction with CD47. While there are no clinical trials on SIRP-1 and SIRP-2 yet, in vitro data showed some promise in maximizing the phagocytic potential of macrophages while avoiding the antigen sink, which dampens the effectiveness of anti-CD47 agents.

### 2.3. SIRPα Fusion Proteins: TTI-621, TTI-622, ALX-148

SIRPα fusion proteins are SIRPα-IgG complexes that may offer an advantage over classic anti-SIRPα mAbs such as BI 765063 and cc-95251 due to dual action via inhibition of SIRPα-CD47 axis and engagement of macrophage Fc, through their IgG counterparts.

The proteins TTI-621 and ALX-148 were created by the fusion of a human SIRPα with an IgG1 Fc portion, while TTI-622 is fused with an IgG4 Fc fragment. The difference between an IgG1 and IgG4 tail is significant; both bind well to FcγRI to deliver a macrophage-activating signal, but IgG1 binds more strongly to FcγRII and FcγRIII than IgG4 does. This may be why TTI-621 enhanced phagocytosis by both M1 and M2 macrophages equally well, whereas SIRPαFc with an IgG4 tail (TTI-622) induced significantly less phagocytosis by M2 macrophages [36]. Both M2 and M1 macrophages are important players in the tumor environment. Still, because M2 macrophages exhibit pro-tumoral effects and M1 macrophages have anti-tumoral effects, tumors generally suppress M1 macrophages leading to their general scarcity in the TME [37]. The ability of TTI-621 to induce prophagocytic responses in M1 macrophages suggests that it may recruit and activate them, more potently inducing tumor phagocytosis than TTI-622. 

Intravenous TTI-621 has been successful and well-tolerated in patients with CTCL, with a 5/19 (26%) ORR in MF and 1/4 (25%) ORR in Sézary syndrome [38]. When TTI-621 was administered intralesionally, the efficacy was higher, with 34% ORR for MF patients [39]. Notably, cohorts receiving 1 mg injection had as good of a response as those receiving 10 mg injections at 1.5 weeks, with CAILS score reductions from a baseline of roughly 50% [39]. Beyond about two weeks of treatment with intralesional injection, improvement was negligible, with about 10% more reduction in CAILS scores across all quantities [39], indicating the pivotal role of that therapy in the innate immune response. 

There were fewer side effects with intralesional injection compared to the IV route. For IV administration, the most common adverse effect was infusion-related reactions (43%) and thrombocytopenia (25%); there were no infusion-related reactions or thrombocytopenia with intralesional injection [39,40]. Side effects unique to intralesional injection were pain at the injection site (31%) and headache (17%) [39]. Shared side effects included: fatigue (43% intralesional vs. 18% IV), chills (34% intralesional vs. 18% IV), and nausea (17% intralesional vs. 12% IV) [39,40]. Compared to IV TTI-621, a low-dose (1 mg) intralesional injection of TTI-621 is more effective and has fewer side effects. The response to intralesional injection occurred quickly, within two weeks, and prolonged treatment did not create a significant response. Thus, the timing for any adjuvant therapy to potentiate the innate immune response or boost the adaptive immune response should be within two weeks of beginning the TTI-621 injections.

Patients who responded to intralesional treatment with TTI-621 were found to have fewer malignant T cells, increased macrophage phagocytosis, and increased infiltration of tumors by NK cells [39]. NK cells are crucial in mediating the effect of TTI-621 since their depletion in a mouse model resulted in attenuation of the anti-tumor effect [15,41]. This suggests that clinical response to anti-CD47 therapy against CTCL relies on the influx of cytotoxic natural killer cells to the TME. To take advantage of this, bispecific antibodies can be engineered to target NK cells via CD16 or other receptors. Potentiating NK cells can result in enhanced Fc-mediated direct tumor killing and assist with ADCC, contributing to the clinical response of targeted treatment of CTCL. It has been performed already as CD16/CD30 bispecific (AFM13), and the efficacy of such a strategy is currently being explored in phase II clinical trial for relapsed/refractory CD30+ T-cell lymphoma or transformed mycosis fungoides (NCT04101331). 

The CTCL-specific strategies in future therapies targeting the CD47-SIRPα axis may go in two directions: agents without an Fc fragment to engage only macrophages and therapies utilizing Fc fragments to boost ADCC and tumor killing by NK cells.

ALX148 is a SIRPα-IgG1 fused with an inactive Fc region [42]. In an ongoing phase 1 clinical trial, ten patients with head and neck squamous cell carcinoma (HNSCC) received pembrolizumab in combination with ALX148. They had an ORR of 40%, which is about twice as effective as those with a checkpoint inhibitor alone [43]. 

Like other mAbs, the most common side effects of TTI-621 and ALX-148 treatment included infusion-related reactions, chills, fatigue, nausea, pyrexia, pruritis, diarrhea, and thrombocytopenia, which were reversible [38]. The minimal erythrocyte binding permits the use of an IgG1-based fusion protein which maximizes macrophage phagocytosis of tumor cells with less concern for the opsonization of RBCs [36]. Additionally, this makes the pharmacokinetic profile of TTI-621 superior as minimal binding to erythrocytes decreases antigen sink due to the strength of the IgG1 binding [36]. 

### 2.4. Small Molecules

Due to their high molecular weights, antibodies have poor tissue penetration, posing an issue for skin cancer. Two small molecules of high-affinity variants of CD47 and SIRPα ectodomains, called Velcro-CD47 and CV1, have been developed to tackle this problem.

Velcro-CD47 is a high-affinity variant of human CD47 ectodomain extended at the N-terminus with a short three-amino-acid peptide. It potently antagonized binding to SIRPα on human macrophages in vitro with similar potency as CV1, which is a truncated SIRPα variant with a greater than 5000-fold affinity to CD47 compared to natural SIRPα [44,45]. However, not many studies have been published on Velcro-CD47 or CV1. These variants demonstrate a pure approach to blocking CD47-SIRPα relative to other therapies. They thus may have been too simplistic for the complex human TME, where more immune effector cells are present. There are more targets for tumor suppression of immune response than an immunosuppressed murine model [46]. Because they are so “pure”, they may also lack specificity, perhaps causing them to bind all myeloid, lymphoid, and erythroid cell populations and raise concerns for side effects such as anemia [45]. Thus, other targeting modes may have more focused anti-tumor activity with fewer side effects.

### 2.5. Bispecific Antibody/Fusion Proteins: DSP107, HX009, IBI322, and SL-172154

Bispecific antibodies are two immunoglobulin chains of differing specificity fused into a single antibody molecule [47]. They are designed to selectively block CD47 in tumor cells that co-express another tumor-specific molecule to achieve higher specificity and avoid RBC toxicity [48]. Because they are two effector molecules combined, they are dual-targeted towards the CD47-SIRPα and another pathway. Additionally, they can effectively cross-link macrophages and cancer cells, bringing them near each other, which is unachievable for a mixture of monospecific antibodies [25,49]. Close physical proximity may, in turn, carry out a new function, and there may be a separate activation pathway that comes from the engagement of two targets simultaneously. It is known that CD3- TCR antigen signaling works this way, as monovalent binding is insufficient to induce antigenic modulation or cytokine release, things that are achievable by crosslinking CD3 to the surface of T cells [49]. Additionally, the conformation of antibodies in bispecific antibodies dramatically affects their effectiveness. T-cell bispecific antibodies elicited more potent antitumor activity in vitro and in vivo when its binding domains were placed in a cis-configuration rather than a trans-configuration, improving cytotoxicity to 2000-fold [50]. Because of this, it may be reasonable to assume that since particular proximity is essential in the biological immune response, a bispecific antibody response may also be necessary.

Bispecific antibodies have good utility, as they are effectively two synergistic agents combined and delivered in one administration. Although the concept is promising, there are challenges in developing bispecific antibodies. There is a lack of detailed studies on the synergy of CD47 with other targets. Their cellular distribution is complex, and cell-mediated cytotoxicity might be inefficient due to suboptimal pairings of their component heavy and light chains [51]. Because of this, developing a successful bispecific antibody relies on matching targets based on synergy, cellular distribution, and Fc-mediated effector functions [51]. Bispecific antibodies currently in clinical trials include IBI322, DSP107, HX009, and SL-172154.

IBI322 is a bispecific antibody of CD47 and PD-L1. PD-L1 is an inhibitory membrane protein expressed in tumors, suppressing T cell-mediated tumor eradication [52]. Thus, IBI322 should preferentially bind tumor cells and target them preferentially. In mice, IBI322 was shown to effectively accumulate in PD-L1 positive tumors and induce complete tumor regression of Raji-hPD-L1 (lymphoblast-like cells) and A375 (malignant melanoma cells), with only marginal RBC depletion [48]. It induced phagocytosis like an anti-CD47 mAb and blocked the binding of PD-1 to PD-L1 like an anti-PD-L1 mAb would, but with higher efficacy of either agent alone or combination [48]. The prospect of this dual effect, along with minimal RBC depletion, makes IBI322 a promising therapy. Phase 1 clinical trials of IBI322 as monotherapy or combination therapy for advanced malignant tumors are in their early stages (NCT04912466) (NCT04328831). 

HX009 is also an anti-CD47 and PD-1 bispecific antibody for which there are preliminary results from a phase 1 clinical trial for patients with advanced solid tumors (NCT04097769). Engagement of PD-L1 with its ligand, PDL1, causes T cell dysfunction, neutralization, and exhaustion [53]. Therefore, PD-1 can block this inhibition signaling, reactivating effector T cells [54]. Of twenty-one patients, six had the best overall response of stable disease, three had partial responses, only one patient had treatment-related anemia, and none suffered thrombocytopenia [55]. Other side effects included nausea, rash, vomiting, and decreased appetite [55]. 

DSP107 binds CD47 and 4-1BB, a costimulatory receptor upregulated upon TCR/MHC interaction to stimulate tumor-reactive T cells [56]. Because 4-1BB activation by soluble 4-1BBL requires cross-linking for stabilization, DSP107 can trigger 4-1BB signaling only after binding to CD47; this CD47-mediated immobilization facilitates the delivery of the 1-1BBL-4-1BB costimulatory signal to local T cells [56]. This allows for effective and powerful innate and adaptive immune responses [56]. Treatment with DSP107 alone and in combination with rituximab triggered significant pro-phagocytic activity against DLBCL cancer lines in vitro, equal to that of both CD47 mAb and SIRPα:Fc [56]. DSP107 also increased T cell cytotoxicity in vitro, and injection of peripheral blood mononuclear cells and DSP107 into mice with SUDHL6 xenografts (DLBCL cell line) significantly reduced tumor size compared to treatment with peripheral blood mononuclear cells alone [56]. A phase 1 clinical trial of DSP107 alone and in combination with atezolizumab for patients with advanced solid tumors is in the recruitment phase (NCT04440735). Because of the high expression of 4-1BB in lymphomas and success in an in vitro model of DLBCL, DSP107 may be promising for hematologic malignancies [57].

SL-172154 is made of a SIRPα-Fc fused to CD40L, with the idea of directing CD47 blockade towards lymphocytes expressing CD40L and minimizing RBC sink. It was studied in a phase 1 trial for patients with platinum-resistant ovarian cancer and found to have a high engagement of CD40+ lymphocytes and minimal RBC binding, but the best response was a stable disease, and only in 29% of patients (4/14; side effects included IRR (93%), fatigue (47%) and nausea (27%) [44]. While CD40+ lymphocytes were highly engaged, treatment with SL-172154 was not as successful as non-specific, non-fusion protein anti-CD47 treatments, which have clinical effects greater than stable disease and about half the amount of side effects. This calls attention to the importance of targeting various portions of TME rather than just lymphocytes, which even become particularly challenging in T-cell lymphoma due to the crossover of target molecules on malignant and bystander T-cells. After leukocytes became activated by CD47 blockade, maybe there were not enough activated phagocytic cells nearby to keep the immune response going. Clinical development for SL-172154 involves combination therapy with known treatments for various cancers.

## 3. Current and Potential Combination with Anti-CD47 Agents for CTCL

### 3.1. Current Biologic Agents Approved for CTCL

**Mogamulizumab**, a humanized IgG1 monoclonal antibody (mAb) that targets CC chemokine receptor 4 (CCR4), is currently approved for patients with MF or Sézary syndrome who failed one prior systemic therapy [58]. Mogamulizumab has an overall response rate of 28% in CTCL [59]. Regulatory T cells (Treg) and type 2 helper T cells (Th2) also express CCR4, so depleting them from the TME helps to expose tumors that have been escaping immune surveillance [60]. However, depleting Tregs also undoes their natural inhibition of the immune system to organisms in the skin, causing immune activation and resulting in a rash that mimics CTCL in 34–63% of patients, which is a limiting factor for its clinical utility [61]. Mogamulizumab caused a high degree of lymphopenia in 81% of subjects [59]. Other common antibody-dependent side effects of mogamulizumab include infusion-related reactions (35%), diarrhea 24%, and fatigue (24%) [59]. These are the same side effects encountered with CD47 blockers, so presumably, a combination of the two would not resolve them; currently, a trial of mogamulizumab in combination with magrolimab is in the recruitment phase for patients with R/R stage IB-IV MF or Sézary syndrome (NCT04541017) [62]. 

**Alemtuzumab** is an IgG1 mAb that targets CD52, which is highly expressed on lymphocytes but also monocytes and dendritic cells [63]. Because of this, severe infectious complications are common but can be avoided at low doses. Clinical trial alemtuzumab was completed including a low-dose cohort: 4 of 14 subjects received 3 mg on day 1, 10 mg on day 3, then 15 mg on alternating days, while the remaining 10 received a low dose of 3 mg on day 1, then 10 mg on alternating days. The overall response rate was 86%, with 21% having a complete response. This high response rate may be since 3 of the 14 subjects had untreated, advanced Sézary syndrome, while the remaining 11 had relapsed/refractory Sézary syndrome. Overall infectious complications occurred in 29% of subjects; all included in the group treated with the relatively higher dose, ultimately receiving 15 mg on alternating days. No hematologic toxicity or infections were seen in the low-dose group. However, there are only anecdotal cases of the usage of alemtuzumab for mycosis fungoides, with mixed results. Delivery to the skin of two agents can be an issue, but the benefit for Sézary syndrome of that combination is apparent. It may require a lower dose of alemtuzumab and preferably subcutaneous administration. The unilateral targeting of the innate response without stimulation of the adaptive response could be a problem. 

**Lacutamab** (IPH4012) is a mAb targeting KIR3DL2, a cell surface protein expressed in CTCL, predominantly in Sézary syndrome [64]. In phase 1 clinical trial with thirty-five subjects with Sézary syndrome, eight subjects with mycosis fungoides, and one subject with primary CTCL, which was not otherwise specified, IPH4012 had an overall response rate was 36%, with a 43% response rate in patients with Sézary syndrome [64]. It also has a favorable side effect profile, which includes peripheral edema (27%) and fatigue (20%) [64]. A phase 2 trial is underway for lacutamab alone and in combination with chemotherapy in patients with advanced CTCL NCT03902184. Similarly to alemtuzumab, most patients with Sézary syndrome may benefit from a combination of lacutamab with anti-CD47 agents.

**Brentuximab vedotin**, an antibody–drug conjugate made of anti-CD30 and monomethyl auristatin E, is approved to treat transformed CD30+ mycosis fungoides and was shown to eliminate bystander cells which highly express CD30 as well [45,65]. It has a high overall response rate of 67% [66]. In a case report, it was shown to be active against CD30- negative mycosis fungoides, suggesting that its effectiveness partly relies on its ability to clear bystander cells from the TME [67]. Brentuximab vedotin may boost the adaptive immune response that can help alleviate the TME of bystander cells, making way for activated macrophages to clear tumors. Thus, combination therapy with brentuximab vedotin and intralesional injection of TTI-621 may work better than ant-CD47 agents delivers intravenously. 

**Pembrolizumab and nivolumab.** Cutaneous lymphoma cells express PD-1 highly: 65% of patients with mycosis fungoides and 89% of patients with Sézary cells are positive for PD-1 [68,69]. There are anti-PD-1 antibodies approved for treating CTCL: nivolumab and pembrolizumab, which bind at different sites of PD-1 [70]. Nivolumab binds the N-terminal loop of PD-1, while pembrolizumab binds the ligand-binding domain of PD-1 [70]. Nivolumab was not very effective for patients with mycosis fungoides, with an ORR of only 15%; however, there was one report of a complete response in a patient with Sézary syndrome [45,71]. Pembrolizumab works better for mycosis fungoides patients with an ORR of 56%, whereas the ORR for Sézary syndrome patients is only 27% [46]. This underscores the significance of the functional differences in binding sites of nivolumab and pembrolizumab, although they exert anti-tumor effects through the exact mechanism of PD-1 signaling blockade.

The PD-1/PD-L1 axis is a negative regulator of the immune response [47]. In a non-disease state, PD-1 positive T cells are recruited to a site of infection, and healthy cells that surround it express PD-L1 to protect themselves from the destruction of the PD-1/PD-L1 interaction, tumor cells cause epigenetic changes within T cells to lead to an exhausted phenotype [47]. PD-1 expression on M2 macrophages has been linked to decreased phagocytic activity as well [48]. Thus, blocking the PD-1/PD-L1 axis may boost macrophages and block tumor cell inhibition signaling, optimizing the immune system against cancer.

Unfortunately, PD-1 blockade is a double-edged sword; in vitro, it reduced the Th2 (pro-tumor) phenotype of non-tumor T cells but enhanced the proliferation of tumor T cells in patients with Sézary syndrome [49]. Perhaps creating a fusion protein by linking an anti-PD-1 antibody with an antibody targeted towards exhausted T cells may tip the balance more towards tumor cell killing over the allowance of proliferation. Because CD47 is also expressed on macrophages, linking anti-PD-1/PD-L1 therapy with anti-CD47/SIRPα as a fusion protein may be beneficial for CTCL treatment, as it has been shown to have good efficacy against melanoma in vitro, even when anti-PD-L1 monotherapy alone was insufficient [26]. 

### 3.2. Current Chemotherapeutic Agents Approved for CTCL

Chemotherapy administered before anti-CD47 therapy may not only be synergistic but also might preserve memory immune response preventing tumor relapse. The benefits of such “neoadjuvant therapy” are multifaceted. Chemotherapy may induce the release of tumor DNA, making its recognition easier. It can also sensitize tumor cells by causing damage to them, inciting their upregulation of “eat me” signals. Lastly, chemotherapy may pre-condition tumors by recruiting more inflammatory cells to infiltrate the tumor, allowing anti-CD47 blockade to work on more cells. Whereas chemotherapy administered after boosting the innate response with anti-CD47 therapy might have a detrimental effect on halting the adaptive immune response.

**Methotrexate** is a folate inhibitor that was used in CTCL for decades. Currently, the utility of methotrexate is limited in CTCL as the second-line medication. At the same time, 5–10 mg of methotrexate weekly can be beneficial to prevent the auto-antibody formations that inevitably happen due to the prolonged use of biologics. 

**Bexarotene** is a synthetic rexinoid used for refractory or persistent early stage CTCL apoptosis. It works by activating retinoid X receptors, causing apoptosis of lymphoma cells. The overall response rate was 45–54% with oral bexarotene. The effect of bexarotene on the immune system is understudied. While early studies showed no impact on Langerhans cells and keratinocytes [50], the recent research demonstrated the stimulation of tissue-resident Treg and anti-inflammatory effect via inhibition of the Th17 axis [51]. A combination of anti-CD47 and bexarotene can be beneficial due to various targets; however, the impact of bexarotene on adaptive immunity remains unclear to predict the long-term outcome of this combination.

**HDAC inhibitors.** Three medications are currently FDA-approved for treating CTCL: vorinostat, belinostat, and romidepsin. Vorinostat is approved for refractory CTCL, and romidepsin and belinostat treat CTCL and peripheral T-cell lymphoma (PTCL). HDAC inhibitors are immunosuppressive [52], raising concern for efficacy alongside anti-CD47 therapy.

**Doxorubicin,** an anthracycline antibiotic, and an aminoglycoside inhibit tumor growth by blocking topo isomerase 2. Doxorubicin is used chiefly as a pegylated formulation for the treatment of CTCL. Its efficacy was established more than 20 years ago, demonstrating OR rate of 80% and a high CR rate of 60% [53]. Unfortunately, relapses are common [54], and doxorubicin is toxic in higher doses. Interestingly, doxorubicin in small amounts is immunostimulatory. Doxorubicin was shown to upregulate CD47 [55] and, while combined with ant-CD47 mAbs, had reduced cardiotoxicity and improved anti-cancer effect [56]. 

**Gemcitabine**, a pyrimidine analog that interferes with cancer DNA production, is efficacious in the treatment of various solid and hematological malignancies. It is also used in the treatment of advanced CTCL, with an ORR of 63% [57]. However, there is conflicting data on its safety profile, with one study citing a high incidence of severe complications such as severe neutropenia (30%), serious infection (26%), and other serious adverse events (26%) including hemolytic uremic syndrome, severe capillary leak syndrome, cardiac arrhythmia leading to acute heart failure, bullous dermatitis, and recurrent influenza-like syndrome at doses of 700–1000 mg/m^3^/day, while another study reports only mild hematologic toxicity and no organ toxicity at 1200 mg/m^3^/day [57,72]. A significant myelosuppression associated with gemcitabine precludes this medication to combine with anti-CD47 agents. 

## 4. Future Targets Currently Not Approved for CTCL in Clinical Trials for Other Cancer and in Preclinical Stage of Development

Continued success in treating patients with CTCL requires the exploration of CD47 antagonists combined with other targets to potentiate both the adaptive and innate immune responses. CD47 blockade can be effective by lowering the threshold for macrophage phagocytosis, while tumor-binding antibodies may be able to direct macrophages against tumors for greater specificity [25]. Coadministration with chemotherapy and agents that clear bystander tumor-infiltrating T cells, activate senescent macrophages, and attract NK cells to the tumor may be highly beneficial.

### 4.1. The Cellular Players in the Tumor Microenvironment of CTCL That Can Be Targeted

The cellular TME of CTCL is composed of dendritic cells, regulatory and tumor-infiltrating lymphocytes, M1 and M2 macrophages, natural killer cells, eosinophils, and fibroblasts (Figure 2). 

The dysregulation of TME leads to impaired immune surveillance contributing to the growth of tumors and cancer progression. The functional overactivation of T-cells in the TME results in T-cell exhaustion. It prevents successful antitumor immunity leading to downregulation of the pro-inflammatory and IFN-γ-related cytokines [73,74]. The depletion of those exhausted cells in other cancers was associated with cytotoxic cell activation. 

The progression of CTCLs is accompanied by a switch from a Th1- predominant cytokine profile to a Th2, suggesting that Th2 cytokines fuel progression [75]. In fact, after co-incubation with Th2 cell cytokines, Sézary cells had increased expression of CD47, which increases their ability to evade phagocytosis [14]. 

Macrophages themselves can also be anti-tumorigenic (M1) or pro-tumorigenic (M2) and are polarized by Th1 and Th2 cells, respectively [76]. Active M1 macrophages, which are PD-1 negative, produce anti-tumorigenic pro-inflammatory cytokines and are relatively scarce in the TME [77]. M1 macrophages can kill tumor cells, inhibit angiogenesis, normalize tumor vessels, and improve adaptive immune responses [78]. The presence of senescent M2 macrophages, which express PD-1, makes CD47 blockade less effective [77]. Thus, the macrophage prophagocytic response to CD47 blockade may be optimized by using a second signal to activate senescent M2 macrophages to an anti-tumor, M1 form. This has been achieved by the addition of CSF-1R inhibitors to SIRPα blocking antibodies [79]. CSF-1R is a tyrosine kinase CSF receptor expressed on macrophages that binds macrophage colony-stimulating factor (MCSF) secreted by cancer cells to drive differentiation to an M2 phenotype; thus, blocking it skews the M1 to M2 ratio towards the M1 lineage and enhances phagocytosis of cancer cells [79]. 

### 4.2. Other Checkpoint Molecules Besides Anti-PD1/PDL1

In addition to PD-1, another immune suppressive checkpoint molecule highly expressed in CTCL dampens T cell response: cytotoxic T lymphocyte-associated antigen 4, or CTLA-4 [74]. CTLA-4 outcompetes CD28 on T cells for its ligands, inhibiting T-cell activation and causing energy. Unfortunately, an anti-CTLA4 mAb (Ipilimumab) was found to have poor blocking activity of CTLA-4 interactions. Any clinical effectiveness was dependent on Fc receptor engagement, making prospects look grim for the future of anti-CTLA-4 mAbs as monotherapy [80]. A combination of anti-PD1 and anti-CTLA4 blockade therapy with nivolumab and ipilimumab was found to have a higher ORR (57%) than monotherapy with ipilimumab (19%) and nivolumab (44%) in a phase 3 study in patients with untreated melanoma, suggesting increased efficacy in combination. Still, whether this increased effectiveness is from Fc engagement or CTLA-4 blocking [81]. 

Immune checkpoints that play roles in immune tolerance of various cancers include T cell immunoglobulin-3 (TIM-3), V-domain Ig-containing suppressor of T cell activation (VISTA), and lymphocyte-activation gene 3 (LAG-3). While they are rarely expressed on T cell lymphomas [82], LAG3 is expressed on CD4+ and CD8+ tumor-infiltrating lymphocytes (Figure 3) and could be a druggable target alone or in combination with CD47. In the case of combination, we predict a synergism due to targeting various parts of TME.

Additionally, while TIM-3 is highly expressed on exhausted T cells in response to chronic viral infection and tumor, it was found that TIM-3 is neither necessary nor sufficient for developing T cell exhaustion [83]. However, it was enough to drive resistance to PD-L1 blockade during chronic infection, suggesting a role for combination therapy to block PD-L/PD-L1 and TIM-3 [83].

### 4.3. TAG-72 

Tumor-associated glycoprotein-72 (TAG-72), an established marker for many tumors, has recently been found to be expressed at a significantly higher level in T cells from T cell lymphoma patients compared to healthy controls and detected at a frequency of 36% among cells in the skin of CTCL patients and all Sezary patients [84]. TAG-72 is a known potential target for CAR-T cells in certain solid tumors and for CTCL. Interestingly, dual CAR-T cells targeting two tumor antigens: TAG-72 (tumor-associated glycoprotein 72) and CD47, were recently generated and showed great benefit for ovarian cancer [85]. It would be interesting to see the applicability of this approach for CTCL patients as well. 

### 4.4. Immature DCs

Immature DCs in the TME of CTCL induce immunotolerance to CTCL, playing a role in its immune escape [86]. Because they are central inducers of the immune response, they may improve anti-cancer immunity in cases where targeting T cells alone is an effective immunotherapy [87]. Treatment modalities targeting DCs under investigation include in vivo expansion with Flt3L [88], in vivo activation with exogenous STING [89] or TLR agonists [90], blockade of inhibitory signals with VEGF [91], and vaccination with dendritic cells [92] or peptides [93].

### 4.5. CD40 Ligand in TME of Lymphoma

CD40 is widely expressed in innate immune cells and when targeted with a CD40 agonist, it can activate APCs, increase MHC and costimulatory molecule expression, and antigen cross-presentation to trigger an adaptive immune response. This makes CD40 an attractive target for the treatment of CTCL as its progression is accompanied by a dumping of the immune response. Recently, an anti-CD40 agonist called ADC-1013 was shown to induce a long-lasting antitumor response and T cell-dependent immunologic memory against bladder cancer [94]. A study on T cell lymphoma showed that a CD40 antagonist also has tumor-killing effects, causing a rapid cytotoxic T cell response independent of helper T cells which expanded by tenfold over five days, eradicating lymphoma in mice [95]. Failure of SL-172154 to produce a sustainable response indicates the importance of targeting other players in TME as well.

### 4.6. BTLA

Exhausted bystander cells express both CD30 and BTLA, and while there has been success targeting CD30 with brentuximab vedotin, no agents are targeting BTLA to our knowledge [41]. Anti-CD47 therapy decreases the density of exhausted BTLA+ CD4+ T cells around malignant T cells [41]. Immune checkpoint blockade of PD-1 aimed toward reinvigorating exhausted T cell effector function has had clinical effectiveness but is limited by the fact that PD-1 blockade increases the proliferation of malignant cells [49]. Thus, optimal utilization of a PD-1 blocker may be accomplished by directing the blockade towards exhausted T cells rather than tumor cells which would then benefit from the blockade. An anti-BTLA and PD-1 bispecific antibody may be able to accomplish this by using BTLA to home in on exhausted T cells, leading to their removal and stimulation of the immune system against tumors.

### 4.7. Mannose

Tumor-associated macrophages are essential components in the TME, and clinical studies targeting tumor-associated macrophages have shown satisfactory outcomes decrease tumor burden and metastasis of many solid tumors [96]. Mannose receptors are highly expressed on M2 macrophages specifically, and M2 macrophages have been successfully targeted in vivo; nanoparticles grafted with a mannose ligand and a BBB-penetrated transferrin receptor binding peptide (TfR-T12) were able to target M2 macrophages and convert them into antitumor M1 macrophages in mouse models of glioma, which inhibited glioma cell proliferation [97]. Mannose may be used to target M2 macrophages similarly against CTCL.

### 4.8. TSP1

Thrombospondin-1 (TSP) is a secreted glycoprotein that is also a ligand of CD47; it is a known gatekeeper of tumor progression that sometimes has opposite effects [98]. Anti-tumor effects of TSP1 include the promotion of M1 macrophage recruitment and cytotoxicity and inhibition of angiogenesis [99]. Pro-tumor effects include the promotion of tumor adhesion and proliferation [98]. TSP1 has also been reported to induce Treg cells in melanoma, and survival was prolonged significantly in mice treated with anti-TSP1 mAb [100]. TSP1 also causes the immunotolerance of the tumor by dendritic cells; silencing TSP-1 expression via small interfering RNA in mouse models had potent anti-tumor effects that also led to an increase in cytotoxic T cells [101]. These pro-tumorigenic effects may be blocked by blocking the TSP-1/CD47 axis, but possibly at the expense of anti-tumor results. This may be especially worth trying in CTCL, as serum TSP1 levels are significantly elevated in patients with mycoses fungoides and even more so in Sézary syndrome, which suggests that the pro-tumorigenic effects of TSP1 expression in CTCL outweigh the anti-tumorigenic effects [102]. 

## 5. Concluding Remarks

Potentiation of the efficacy of immunotherapy for CTCL can be improved by addressing poor tumor antigenicity, and inactive status of tumor-infiltrated lymphocytes, highlighting a need for synergistic approaches to boost antitumor immunity that tackles the immunosuppressive TME as well. Novel therapeutic approaches include bispecific antibodies of various designs and tetrameric peptide-MHC-class I complexes that developed to engage multiple T cell receptors on the surface of a T cell, allowing for increased avidity to TCRs [103]. Tetramers are already used in personalized cancer treatment; they isolate antigen-specific T cells for adoptive T cell therapy. A patient’s T cells are isolated, cloned, expanded to enhance anti-tumor activity, and then transferred back to them. Because the cancer cells in question in CTCL are T cells, the future of CTCL therapy may include tetramers designed to engage antigens specific to tumor cells or toward exhausted T cells, which may then be used for adoptive T cell therapy.

## Figures and Tables

**Figure 1 cells-11-03591-f001:**
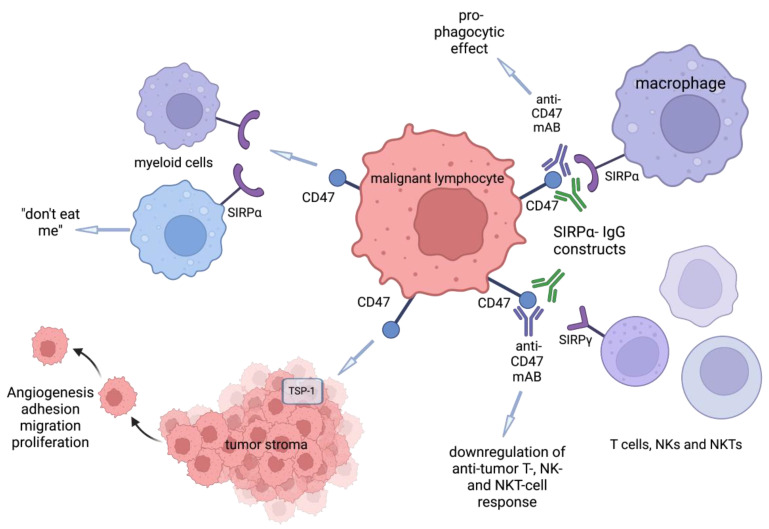
Anti-CD47 mAb boosts tumor phagocytosis by macrophages but dampens the anti-tumor T-cell response due to nonspecific blockage of SIRPα and SIRPγ.

**Figure 2 cells-11-03591-f002:**
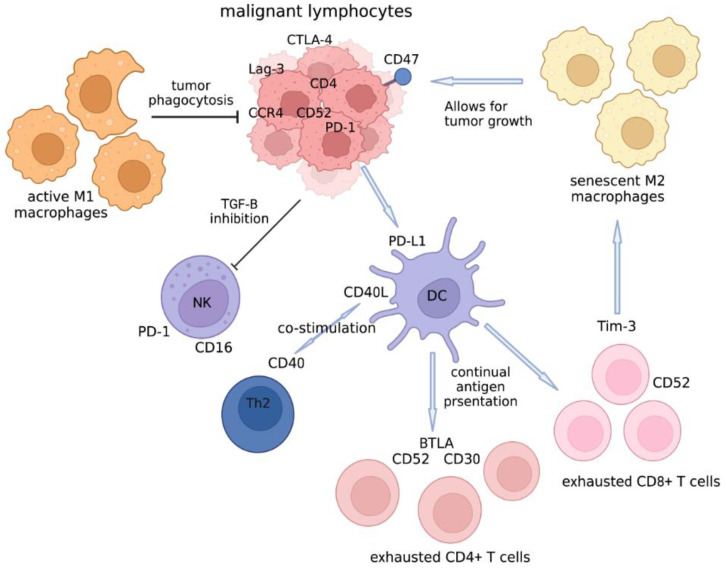
The tumor microenvironment of CTCL.

**Figure 3 cells-11-03591-f003:**
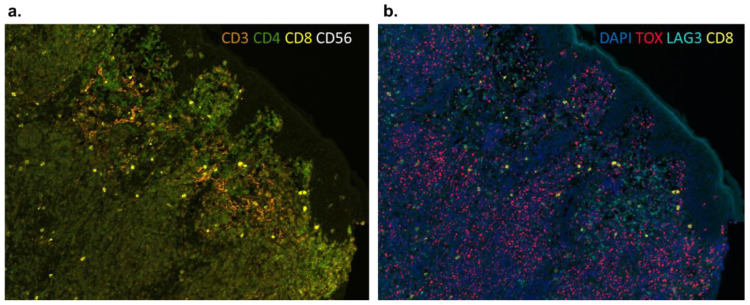
LAG3 expression in the skin of a patient with Sezary syndrome. (**a**) Dense lymphocytic infiltrate of the skin by CD3^dim^ CD4+ Sezary cells. Note CD4+ tumor-infiltrating lymphocytes in the top portion of the reticular dermis. (**b**) Moderate expression of LAG3 on CD4+ and CD8+ tumor-infiltrating lymphocytes but not on TOX+ Sezary cells.

**Table 1 cells-11-03591-t001:** Anti-CD47 monoclonal antibodies.

Company	FortySeven/Gilead	Arch Oncology		I-MAB Biopharma	Cellgene	Surface Oncology	Jiangsu HengRui Medicine
Candidate	Margolimab (5F9)	AO-176	Ti-061	TJ011133 (TJC4)	CC-90002	SRF231	SHR 1603
Fc isotype	IgG4	IgG2	IgG4	IgG4	IgG4-PE	IgG4	IgG4
Lead indication	MDS/AML; DLBL; Solid tumors; Colorectal CA; Hematologic malignancies	Solid tumors; MM; Preclinical: lymphoma and TLL	Solid tumors	R/R solid tumors and lymphoma	Not been used as monotherapy R/R NHL in combination	B cell lymphoma, R/R solid tumors	Advanced CA; hematologic malignancies

Abbreviations: mAb, monoclonal antibody; IgG, immunoglobulin; WT, wild type; CA, cancer; HNSCC, head and neck squamous cell carcinoma; NHL, non-Hodgkin lymphoma; MDS, myelodysplastic syndrome; AML, acute myeloid leukemia; DLBL, diffuse large B-cell lymphoma; NSCLC, non-small cell lung cancer; SCC, squamous cell carcinoma.

**Table 2 cells-11-03591-t002:** SIRPα, SIRPα proteins, and bispecific antibodies.

	SIRPα Antibody		SIRPα Proteins				Bispecific Antibody			
Company	Celgene	OSE Immunotherapeutic	Weissman’s group	ALX Oncology	Trillium Therapeutics		Kahr Medical	Waterstone Hanxbio Pty Ltd. (Wuhan, China)	Invent Biologics	Shattuck Labs
Candidate	CC-95251	BI 765063 (OSE-172)	CV1	ALX-148	TTI-621	TTI-622	DSP107	HX009	IBI322	SL-172154
Molecule	mAb	mAb IgG4	Truncated SIRPα protein	WT SIRPα-IgG1 fusion with inactive Fc	WT SIRPα-IgG1 Fc fusion	WT SIRPα-IgG4 Fc fusion	SIRPα/41BB	CD47/PD1	CD47/PDL1	SIRPα/40L
Lead indication	Solids tumor, leukemia/lymphoma	Advanced solid tumors	Lymphoma; breast CA	HNSCC, gastric CA, breast CA, NHL, MDS, AML	Hematologic malignancies		NSCLC, SCC, advanced solid tumors	Advanced solid tumors	NSCLC, cervical, esophageal, and liver CA, HNSCC	Ovarian CA

Abbreviations: mAb, monoclonal antibody; Ig, immunoglobulin; WT, wild type; CA, cancer; HNSCC, head and neck squamous cell carcinoma; NHL, non-Hodgkin lymphoma; MDS, myelodysplastic syndrome; AML, acute myeloid leukemia; NSCLC, non-small cell lung cancer; SCC, squamous cell carcinoma.

## Data Availability

Not applicable.

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
