# Peer review of "Targeting the CD47-SIRPα Axis: Present Therapies and the Future for Cutaneous T-cell Lymphoma"

_cells, 2022, doi:10.3390/cells11223591_

Round 1

Reviewer 1 Report

The authors summarize the novel treatment options of CTCL, focusing on various immunotherapy options. The review puts a special emphasis on the CD47-SIRPa axis. All immunotherapies available and being investigated are listed as well as possible future therapeutic options. 

The work is well organized, however quite dense and requires good immunological background to fully understand all parts.

No major issues were found. 

Only a minor comment on Table 1. 

CA is used for cancer and it is used as hematological CA, however in Table 2 Hematological malignancies are quoted. I recommend changing the hematological Ca to hematological malignancies to have the same, more acceptable terminology.

Author Response

Thank you very much for the opportunity to resubmit our manuscript titled “Targeting the CD47-SIRPα axis: present therapies and the future for cutaneous T-cell lymphoma.” We appreciate the encouraging comments. We have edited the manuscript as recommended and believe that the manuscript has improved significantly. Please find the manuscript with highlighted changes attached. 

The point-by-pointed answers are presented below.

Reviewer 1

The authors summarize the novel treatment options of CTCL, focusing on various immunotherapy options. The review puts a special emphasis on the CD47-SIRPa axis. All immunotherapies available and being investigated are listed as well as possible future therapeutic options.  The work is well organized, however quite dense, and requires good immunological background to fully understand all parts.

No major issues were found. 

A: We thank reviewer 1 for the kind evaluation of our work.

Only a minor comment on Table 1. 

CA is used for cancer and it is used as hematological CA, however in Table 2 Hematological malignancies are quoted. I recommend changing the hematological Ca to hematological malignancies to have the same, more acceptable terminology

A: We have corrected as recommended.

Reviewer 2 Report

The authors reviewed comprehensive data about targeting CD47-SIRPα axis in cutaneous T-cell lymphoma. Some details will need to be clarified and mentioned:

(1) The title of the manuscript is "Targeting the CD47-SIRPα axis: present therapies and the future for cutaneous T-cell lymphoma". However, many drug targeting CD47-SIRPα in the manuscript were not studied for cutaneous T-cell lymphoma. The drug studies about cutaneous T-cell lymphoma or other malignancies should be listed separately.

(2) Some details about drug studies, such as patient numbers, and overall response rates could be listed in Tables. In addition, the side effects of these study drugs were not mentioned in this review. 

(3)  The clinical trials to combine drugs targeting CD47-SIRPα and other agents should be listed in one table and mention the numbers on ClinicalTrial.gov if needed. 

Author Response

Thank you very much for the opportunity to resubmit our manuscript titled “Targeting the CD47-SIRPα axis: present therapies and the future for cutaneous T-cell lymphoma.” We appreciate the encouraging comments. We have edited the manuscript as recommended and believe that the manuscript has improved significantly. Please find the manuscript with highlighted changes attached. 

The point-by-pointed answers are presented below.

Reviewer 2

The authors reviewed comprehensive data about targeting the CD47-SIRPα axis in cutaneous T-cell lymphoma. Some details will need to be clarified and mentioned:

(1) The title of the manuscript is "Targeting the CD47-SIRPα axis: present therapies and the future for cutaneous T-cell lymphoma". However, many drug targeting CD47-SIRPα in the manuscript were not studied for cutaneous T-cell lymphoma. The drug studies about cutaneous T-cell lymphoma or other malignancies should be listed separately.

A: The purpose of our study was to provide a comprehensive vision of all anti-CD47 agents currently in development and examine how they can be useful in the CTCL field. Only two drugs (margolimab and TTI-621) have been used in CTCL so far, and none of them are FDA approved for CTCL. Presenting just those two is pointless because the senior author has already published the results of the clinical trials with TTI-621. We have composed the medications in our review by chemical structure and believe this is a correct and comprehensive approach to understanding where the field of anti-CD47 therapy is moving.

(2) Some details about drug studies, such as patient numbers, and overall response rates could be listed in Tables. In addition, the side effects of these study drugs were not mentioned in this review.

A: Since most of those drugs were not used in CTCL, providing the number of patients for every single cancer (most of the studies were open for enrollment of any cancers and enrolled literally one of each kind) would be very overwhelming, not adding anything to the main idea.

We have discussed side effects such as “drug sink” leading to thrombocytopenia, anemia, and a decrease in other CD47-expressing cells. We have discussed infusion-related reactions extensively. This is pretty much the whole specter of the side effects. As a class, the medications are very well tolerated.   

(3)  The clinical trials to combine drugs targeting CD47-SIRPα and other agents should be listed in one table and mention the numbers on ClinicalTrial.gov if needed

A: Due to space limitations, we limited the data only to the name of the drugs.

Round 2

Reviewer 2 Report

The authors have addressed to question from my comments